

# HRLT: A high-resolution (1 day, 1 km) and long-term
# (1961–2019) gridded dataset for temperature and
# precipitation across China
Rongzhu Qin, Zeyu Zhao, Jia Xu, Jian-Sheng Ye, Feng-Min Li, Feng Zhang[*]
State Key Laboratory of Grassland Agro-ecosystems, College of Ecology, Lanzhou University,
Lanzhou, 730000, China
[*] Corresponding author: Feng Zhang
Tel.: +86 13919274617
Fax: +86 09318912561
E-mail: zhangfeng@lzu.edu.cn
Address: College of Ecology, Lanzhou University, 222 Tian Shui South Road, Lanzhou, 730000,
China



## Abstract

Accurate long-term temperature and precipitation estimates at high spatial and temporal resolutions are vital for a wide variety of climatological studies. We have produced a new, publicly available, daily, gridded maximum temperature, minimum temperature, and precipitation dataset for China with a high spatial resolution of 1 km and over a long-term period (1961 to 2019). It has been named the HRLT and the dataset is publicly available at https://doi.org/10.1594/PANGAEA.941329 (Qin and Zhang, 2022). In this study, the daily gridded data were interpolated using comprehensive statistical analyses, which included machine learning, the generalized additive model, and thin plate splines. It is based on the 0.5° × 0.5° grid dataset from the China Meteorological Administration, together with covariates for elevation, aspect, slope, topographic wetness index, latitude, and longitude. The accuracy of the HRLT daily dataset was assessed using observation data from meteorological stations across China. The maximum and minimum temperature estimates were more accurate than the precipitation estimates. For maximum temperature, the mean absolute error (MAE), root mean square error (RMSE), Pearson's correlation coefficient (Cor), coefficient of determination after adjustment ($R^2$), and Nash-Sutcliffe modeling efficiency (NSE) were 1.07 ℃, 1.62 ℃, 0.99, 0.98, and 0.98, respectively. For minimum temperature, the MAE, RMSE, Cor, $R^2$, and NSE were 1.08 ℃, 1.53 ℃, 0.99, 0.99, and 0.99, respectively. For precipitation, the MAE, RMSE, Cor, $R^2$, and NSE were 1.30 mm, 4.78 mm, 0.84, 0.71, and 0.70, respectively. The accuracy of the HRLT was compared to those of the other three existing datasets and its accuracy was either greater than the others, especially for precipitation, or comparable in accuracy, but with higher spatial resolution or over a longer time period. In summary, the HRLT dataset, which has a high spatial resolution, covers a longer period of time and has reliable accuracy, is suitable for future



environmental analyses, especially the effects of extreme weather.

# 1 Introduction

Climate change has led to an increase in the frequency and severity of extreme temperature
and precipitation events (Myhre et al., 2019), and these events have affected vegetation growth (Xu
et al., 2019), especially crop growth (Rao et al., 2015; Li et al., 2019b; Lu et al., 2018; Lobell et al.,
2011; Lesk et al., 2016). Thus, long-term and accurate daily maximum temperature, minimum
temperature, and precipitation data are important when attempting to reveal the mechanism
underlying the effects of extreme climate on plants, predicting disasters (such as drought, frost, and
floods), and for agricultural and forestry management. Although the meteorological observation
network makes better use of the data from meteorological stations (Merino et al., 2014; Yang et al.,
2014), there is a tradeoff between large spatial scale and the high density of stations in the
meteorological observation network. Moreover, the installation and maintenance of meteorological
stations are challenging in harsh areas (Hartl et al., 2020). Daily and gridded meteorological datasets
are also essential inputs for many models related to terrestrial, hydrological, and ecological systems
(Iizumi et al., 2017; Wang et al., 2018; Zhang et al., 2018; Lee et al., 2019). High-resolution, long-
term, and accurate gridded datasets can help improve the performance of these models.
Researchers have previously used interpolation methods, such as inverse distance weighting,
kriging, and regression analysis, to produce grid meteorological data (Brinckmann et al., 2016;
Herrera et al., 2019; Schamm et al., 2014). However, the accuracy of these interpolation results is
limited by the density of the meteorological stations. In recent years, artificial intelligence, machine
learning methods, such as random forest (Chen et al., 2021; Sekulić et al., 2021); artificial neural



networks (Sadeghi et al., 2021), and support vector machines (He et al., 2021) have been gradually
and widely applied to meteorological data estimation. Therefore, comprehensive statistical analyses
using machine learning and traditional interpolation, such as thin-plate-smoothing splines, are
feasible and reliable methods that can be used to estimate meteorological data.

At present, only a few research institutes in China are developing meteorological datasets for

temperature and precipitation with high spatial and temporal resolutions. Among them, Beijing
Normal University has produced meteorological datasets for 1958–2010 with a resolution of 1 km,
but the latest data is not available (Li et al., 2014). The China Meteorological Administration is also
developing the CMA Land Data Assimilation System product (Shi et al., 2011) and Tsinghua
University has published a driving dataset from 1979 to 2018 with a resolution of 0.1° over China
(He et al., 2020).

We present a new high-resolution daily gridded maximum temperature, minimum temperature,

and precipitation dataset for China (HRLT) with a spatial resolution of 1 × 1 km for the period 1961
to 2019. We created the HRLT dataset using comprehensive statistical analyses, which included
machine learning, the generalized additive model and thin plate splines. It uses the 0.5° × 0.5° grid
dataset from the China Meteorological Administration (CMA) as input data together with other
covariates, including elevation, aspect, slope, topographic wetness index (TWI), latitude, and
longitude. The dataset was created in three steps: (1) preparation of input data and covariates; (2)
the creation of the gridded dataset using comprehensive statistical analyses; and (3) an evaluation
of the accuracy of the gridded dataset and accuracy comparison with other three exiting products
that use meteorological station data.



## 2 Data

### 2.1 The CMA dataset and meteorological stations data

The CMA dataset, which includes the daily surface temperature 0.5° × 0.5° grid dataset (http://101.200.76.197/data/cdcdetail/dataCode/SURF_CLI_CHN_TEM_DAY_GRID_0.5.html) and the daily precipitation 0.5° × 0.5° grid dataset for China (V2.0) (http://101.200.76.197/data/cdcdetail/dataCode/SURF_CLI_CHN_PRE_DAY_GRID_0.5.html), was obtained from the China Meteorological Data Service Centre and was used as the basic input data. The researchers also reported daily precipitation 0.5° × 0.5° grid dataset during 1961-2010 from CAM dataset (Zhao and Zhu, 2015). The daily dataset of surface climatological data for China (V3.0) (http://101.200.76.197/data/cdcdetail/dataCode/SURF_CLI_CHN_MUL_DAY_V3.0.html), which includes 699 meteorological stations, was also obtained from the China Meteorological Data Service Centre and was used to evaluate the new dataset (Fig. 1).

### 2.2 Topographic data

The basic topographic data, including elevation, flow direction, and flow accumulation with a 30 second (approximately 1 km) resolution, were obtained from the HydroSHEDS database. More detailed information can be found at these links: http://www.worldwildlife.org/hydrosheds for general information and http://hydrosheds.cr.usgs.gov for data download and technical information. The "Aspect" and "Slope" option of the Spatial Analyst Tools in ArcGIS10.6 were used to calculate aspect and slope. The specific catchment area (SCA) was calculated based on flow direction and flow accumulation. The TWI is formulated as TWI = ln(SCA / tan(Slope)).



### 2.3 Other datasets


Three temperature and precipitation products with daily resolutions were evaluated using
observed meteorological stations data and the evaluation results were compared to the HRLT
dataset in this study. The China Meteorological Administration Land Data Assimilation System
(CLDAS) version 2 dataset was provided by the China Meteorological Data Service Centre
(https://data.cma.cn/) for 2017 to 2019 with a 0.0625° (approximately 7.5 km) spatial resolution
and a 1 day temporal resolution. The China Meteorological Forcing Dataset (CMFD) (He et al.,
2020; Yang and He, 2019) was obtained from the National Tibetan Plateau Third Pole
Environment Data Center (https://data.tpdc.ac.cn/) for 1979 to 2018 with a spatial resolution of
0.1° (approximately 12 km) and a temporal resolution of 1 day. The historical dataset relating to
the Inter-Sectoral Impact Model Intercomparison Project (ISIMIP3a) was obtained from the web
(https://data.isimip.org/) for 1961 to 2016 with a spatial resolution of 0.5° (approximately 60 km)
and a temporal resolution of 1 day. The daily maximum temperature, minimum temperature, and
precipitation data in the CLDAS and ISIMIP3a were used for evaluation and comparison. The
daily average temperature and precipitation data from the CMFD was also used for evaluation and
comparison.

# 3 Methods


### 3.1 The input data and covariates


In this study, the input data (dependent variable) was the daily 0.5° × 0.5° CMA dataset, which
includes daily maximum temperature, minimum temperature and precipitation. Other covariates
(independent variables) included elevation, aspect, slope, TWI (with a spatial resolution of 1 km),





latitude, and longitude.
**3.2 The interpolation scheme**
As shown in Figure 2, the different combinations of six algorithms, which are the boosted
regression trees (BRT), random forests (RF), neural networks (NN), multivariate adaptive
regression splines (MAR), support vector machines (SVM) and the generalized additive model
(GAM), to predict the input data. Firstly, through k-fold cross validation (k = 10), the input data was
was randomly divided into 10 sub-training datasets and sub-testing datasets. Each algorithm runs in
a loop through all the sub-training sets and calculates the residuals from the sub-testing sets. The
residuals obtained in each loop are retained. The residual of each algorithm is assigned a weight of
0-1 and summed up, and the ensemble of models that has the lowest residual sum is chosen. After
determining the best ensemble of models, surface results were interpolated using the best ensemble
of models, input data and covariates. The thin-plate-smoothing splines (TPS) is used to correct
residual error from the ensemble of models. Therefore, residuals of the ensemble are calculated from
the input data and these values are interpolated using TPS. Surface results from the ensemble add
residuals from the thin-plate-smoothing splines to get the surface result of final model. Compare $R^2$
of surface result from the ensemble and final model, and retain the surface result with higher $R^2$.
**3.3 The methods**
The introduction of individual algorithm (method) and the implementations for model training
(R packages and functions) of that is as follows. After the model training, the function 'predict' in
R package 'raster' used to spatial interpolation for BRT, RF, NN, MAR, SVM and GAM model, and
the function 'interpolate' in R package 'raster' used to spatial interpolation for TPS. More details on



R packages and functions could refer the web (https://www.rdocumentation.org/).

**3.3.1 The BRT model**

As a powerful tool for exploratory regression analysis, BRT is a combination of two techniques:
decision trees and boosting method (Elith et al., 2008). The BRT can automatically detect the best
fit and is robust to missing values and outliers, therefore, BRT now widely used in Remote sensing,
species distribution and meteorological interpolation (Pouteau et al., 2011; Appelhans et al., 2015;
Froeschke and Froeschke, 2011). There are two important parameters in BRT, (1) the tree
complexity (TC): this controls the number of splits in each tree. (2) learning rate (LR): this
determines the contribution of each tree to the growth model. The smaller value of LR, the more
trees will be built. These two parameters together determine the number of trees required for the
best prediction in order to find the combination of parameters that leads to the least prediction error.
The function 'gbm.step' in R package 'dismo' for the BRT implementation. The the tree complexity
was set at 5, the learning rate was set at 0.001. In addition, the 'bag.fraction', which specifies the
proportion of data to be selected at each step, was set at 0.5 and other parameters are default values
in 'gbm.step'.

**3.3.2 The RF model**

Like BRT, the main technology of RF also includes decision trees, however, the way in which
the data to build the trees is selected is different (boosting method for BRT, bagging method for RF).
For regression analysis, the bagging method, which take a random subset of all data for each new
tree that is built, makes the final output based on average of multiple trees (Breiman, 2001). As one
of the most accurate algorithms, RF has been used widely for predicting spatio-temporal variables,
such as temperature and precipitation (He et al., 2016; Mital et al., 2020; Webb et al., 2016). The





function 'randomForest' in R package 'randomForest' for the RF implementation. The importance
was set TRUE, and other parameters are default values in 'randomForest'.
**3.3.3 The NN model**
As a powerful set of tools for solving problems in pattern recognition, data processing, and
non-linear control (Bishop, 1994), the NN consists of a large number of nodes and connections and
it includes input layer, hidden layer and output layer (Lek and Guégan, 1999). Information from
each node in the input layer is fed to the hidden layer. Connections between input layer nodes and
hidden layer nodes can all be given specific weights according to their importance. The connection
between the hidden layer and the output layer is also weighted, so the output is the result of the
weighted sum of the hidden nodes. Information transfer between hidden layer and output layer
through transfer function. Since the 1980s, the NN has been used in a number of fields, such as
prediction for meteorological variables (Snell et al., 2000; Lek and Guégan, 1999; Tang et al., 2020).
The function 'nnet' in R package 'nnet' for the NN implementation. The number of units in the
hidden layer (size) was set 10, the transfer function is linear for the output layer (linout was set
TRUE), the maximum number of iterations (maxit) was set 10000, and other parameters are default
values in 'nnet'.
**3.3.4 The MAR model**
The MAR is an extension of linear model, which can build multiple linear regression models
within the range of predictive variable values by partitioning data (Friedman, 1991; Friedman and
Roosen, 1995). The MAR consists of two steps: firstly, it creates a set of so-called basis functions.
In this process, the range of predictive variable values is divided into several groups. For each group,
separate linear regression was modeled. Secondly, MAR estimates a least square model with its



basis function as the independent variable. Overfitting is avoided by iterating to remove the basis
functions that contribute least to the model fitting. The MAR works well with a large number of
predictor variables, automatically detects interactions between variables and is robust to outliers,
therefore, studies has done on downscaling or predicting meteorological data using MAR (Panda et
al., 2022; Li et al., 2019a; Zawadzka et al., 2020). The function 'earth' in R package 'earth' for the
MAR implementation. Use linear model to estimate standard deviation as a function of the predicted
response (varmod.method = 'lm'). The nfold was set 10, the ncross was set 30, and other parameters
are default values in 'earth'.

**3.3.5 The SVM model**


The SVM is also one of the machine learning supervised algorithms and mainly deals with the

ideas of classification and regression (Vapnik, 1999; Vapnik, 1991; Brereton and Lloyd, 2010). The
SVM is well supported by mathematical theory and can use kernel tricks to efficiently process non-
linear data. With the development of SVM, it also has been widely used in the regression and
prediction of meteorological variables (Belaid and Mellit, 2016; Chen et al., 2010; Tripathi et al.,
2006). In this study, the function 'ksvm' in R package 'kernlab' for the SVM implementation and
all parameters are default values in 'ksvm'.

**3.3.6 The GAM model**


The GAM is an extension of the generalized linear model (GLM). Like GLM, GAM consists

of three important components: the probability distribution of the dependent variable, the linear
predictor and the link function, however, in GAM, the coefficient of the independent variable in the
linear is replaced by a sum of smooth functions (Hastie and Tibshirani, 2017; Liu, 2008). Because
the GAM can deal with nonlinear and non-monotone relationships between dependent and



independent variables, it has been used to predict and interpolate meteorological data (Hjort et al.,
2016; Burnett and Anderson, 2019; Aalto et al., 2013). The function 'gam' in R package 'mgcv' for
the GAM implementation and all parameters are default values in 'gam'.
**3.3.7 The TPS method**
As a traditional interpolation method, the TPS has been widely used to spatially interpolate
surface climate data (Gong et al., 2022; Hancock and Hutchinson, 2006; Risk and James, 2022). In
this study, it used to correct residual error from the ensemble of models. The function 'Tps' in R
package 'fields' for the TPS implementation. The matrix of independent variables consists latitude
and longitude, the vector of dependent variables is residual error in the combinations of above
algorithms, and other parameters are default values in 'Tps'.
**3.4 The interpolation implementation**
A complete operation was constructed per day per variable, so there were 64647 operations
(21549 days × 3 variables) from January 1, 1961 to December 31, 2019 for maximum temperature,
minimum temperature and precipitation. A complete operation for a day per variable requires a
Central Processing Unit core, 18 G of operating memory, and 2 hours of time. In order to shorten
the running time, we carried out parallel computing on a supercomputer platform. Spatial
interpolation work was executed by R version 4.0.2 (R Core Team, 2018) and the R package
"machisplin" (Brown, 2019) was referenced to achieve it.
**3.5 Evaluation metrics**
The mean absolute error (MAE), root mean square error (RMSE), Pearson's correlation
coefficient (Cor), coefficient of determination after adjustment ($R^2$), and Nash-Sutcliffe modeling



efficiency (NSE) were used to evaluate the interpolation results. Pearson's correlation coefficient
was used to evaluate the correlation between the simulated and observed values and the other
metrics are defined separately as follows:

$$MAE = \frac{1}{n}\sum_{i=1}^{n} |S_i - O_i| \tag{1}$$

$$RMSE = \sqrt{\frac{1}{n}\sum_{i=1}^{n}(S_i - O_i)^2} \tag{2}$$

$$R^2 = 1 - (1 - \frac{\sum_{i=1}^{n}(S_i - \bar{O})^2}{\sum_{i=1}^{n}(O_i - \bar{O})^2})\frac{(n-1)}{(n-k-1)} \tag{3}$$

$$NSE = 1 - \frac{\sum_{i=1}^{n}(S_i - O_i)^2}{\sum_{i=1}^{n}(O_i - \bar{O})^2} \tag{4}$$

where $S_i$ and $O_i$ are the model predicted and the experimentally observed values, respectively;
$\bar{O}$ is the mean of the observed values; $n$ is the number of observations; and $k$ is the value of the
independent variable. High Cor, $R^2$, and NSE values, and small RMSE and MAE values indicate
the strength of agreement between the predicted and observed values.

## 4 Results and discussion

### 4.1 Validation of temperature and precipitation

The spatial interpolation results, including daily maximum temperature, minimum temperature,
and precipitation, were validated using meteorological station data. The results of the validation
showed that the daily maximum and minimum temperatures were highly accurate (Fig. 3 and Table
1). The fitting slopes between the simulated and observed values were both close to 1 and the
coefficients of determination after adjustment were 0.98 and 0.99, respectively, for daily maximum
and minimum temperature (Figs. 3a, b). As shown in Table 1, the MAE was 1.07 °C and 1.08 °C,



and the RMSE was 1.62 °C and 1.53 °C for daily maximum and minimum temperatures, respectively.
In addition, the Cor and NSE values were close to 1 for both the daily maximum and minimum
temperatures. Daily precipitation was less accurate than temperature with an $R^2$ of 0.71 (Fig. 3c),
which was mainly caused by underestimating high daily precipitation. However, most of the points
were concentrated in the low daily precipitation section. Furthermore, the MAE and RMSE for daily
precipitation were 1.30 mm and 4.78 mm, respectively; the Cor between the simulated and observed
daily precipitation was 0.84, and the NSE was 0.70 (Table 1).
The interpolation accuracy shows spatial differences (Fig. 4). The $R^2$ values of the daily
maximum and minimum temperatures in southwest China were less than 0.94 and lower than those
for other regions (Figs. 4a, c). The mean absolute errors for the daily maximum and minimum
temperature ranges at most meteorological stations were less than 1 °C. However, there were some
meteorological stations with mean absolute errors of more than 2 °C and these were evenly
distributed across China (Figs. 4b, d). The $R^2$ value for daily precipitation at most meteorological
stations was greater than 0.7 and the MAE decreased from south to north across China (Figs. 4e, f).
The meteorological stations were divided into the middle and lower reaches of the Yangtze
River (MLYR), North China (NC), Northeast China (NEC), Northwest China (NWC), South China
(SC), and Southwest China (SWC) (Fig. 1) according to their diverse geographic and climatic
conditions and administrative areas. The density distribution curve trend for the simulated value and
the observed value was always similar for daily maximum temperature, minimum temperature, and
precipitation in the six regions. The daily maximum and minimum temperatures were all
underestimated in the MLYR, NEC, NWC, SC, and SWC, and the daily minimum temperatures





were all underestimated in the MLYR, NWC, SC, and SWC (Fig. 5). For both daily maximum and
minimum temperatures, the lowest difference between the simulated and observed average values
occurred in NEC, while the greatest difference occurred in SWC (Fig. 5). Except in the NWC region,
the simulated average for daily precipitation was lower than the observed average in the other
regions. The largest difference between simulated and observed averages for daily precipitation
occurred in the SC region, with a value of 0.5 mm (Fig. 5).

Figure 6 shows that the average diurnal variation values for daily temperature and precipitation

based on the meteorological station data were almost the same as our estimations. Compared to the
observations from the meteorological stations, the average values for daily maximum temperature
decreased from 17.79 ℃ to 17.44 ℃ (1.9%) and the average values for daily minimum temperature
decreased from 7.24 ℃ to 6.94 ℃ (4.1%) after interpolation, between 1961 and 2019 (Figs. 6a, b).
The maximum values for daily maximum and minimum temperature measured by the
meteorological stations were 33.35 ℃ and 22.24 ℃, and the minimum values for those were −4.710 ℃
and −14.54 ℃, respectively. After interpolation, these corresponding values became 33.23 ℃ and
22.45 ℃, −5.06 ℃ and −15.01 ℃, respectively. Compared to the observations from meteorological
stations, the average values for daily precipitation decreased from 2.43 mm to 2.31 mm (4.9 %) after
interpolation, between 1961 and 2019 (Fig. 6c).
**4.2 Temporal and spatial distributions of temperature and precipitation**

The results showed that detailed spatial changes in temperature and precipitation over time

could be obtained (Fig. 7). For example, the increase in annual average values (both maximum
temperature and minimum temperature) were obvious over the Tibetan Plateau from 1965 to 2010



(Figs. 7a–h, the d1 and h1 subregions). In addition, compared with other years, the annual average
daily minimum temperature clearly increased in some areas of NWC (Figs. 7e–h, the h2 and h3
subregions) and MLYR (Figs. 7e–h, the h4 subregion) in 2010. The most significant annual
precipitation changes occurred in NEC (Figs. 7i–l, the l1 subregion) between 1965 and 2010.

The distributions of annual average daily maximum and minimum temperatures and annual

precipitation across the six regions of China in 1965, 1980, 1995, and 2010 were analyzed (Fig. 8).
Compared with other years, the areas with smaller values for annual average daily maximum
temperature (less than 0) and annual average daily minimum temperature (less than −10) in SWC
and NWC decreased in 2010 (Figs. 8a1, 8a2, 8b1, 8b2). These areas are mainly distributed on the
Qinghai-Tibet Plateau, which has seen a large increase in temperature over the past few decades.
The density distribution peak for the annual average daily maximum and minimum temperatures in
NEC moved to the right from 1965 to 1995, but moved to the left in 2010 (Figs. 8a3, 8b3). The
mean annual average daily minimum temperature in 2010 was higher in the MLYR, NC, and SC
than in the other three years (Figs. 8b4–6). There was an increase in mean annual precipitation in
the northern part of China over the period 1965–2010 (Figs. 8c2–4). It increased from 335 mm to
415 mm across NWC (Fig. 8c2), from 487 mm to 593 mm across NEC (Fig. 8c3), and from 531
mm to 654 mm across NC (Fig. 8c4). In the MLYR, there were more areas with annual precipitation
of less than 1000 mm, and areas with an annual precipitation of more than 2000 mm increased in
1995 and 2010 compared 1965 and 1980 (Fig. 8c5). Similarly, compared with other years, there
were more areas with annual precipitation of less than 1000 mm and more than 2000 mm in SC in
2010 (Fig. 8c6).

## 4.3 Accuracy comparison with other products


The performances of the CMFD, CLDAS and ISIMIP3a generated daily temperatures and
precipitations were evaluated against observations from all the meteorological stations and
compared their performance with that of our dataset (Figs. 9–11; Tables 2–4). The fitting slopes
between the simulated and observed daily temperature values were always close to 1 for all datasets
(Figs. 9a–c; Figs. 10a–d; Figs. 11a–d). The $R^2$ for the CMFD daily average temperature was
slightly smaller than that for daily minimum temperature in our dataset (Figs. 9b, c), but was equal
to our data set for daily maximum temperature (Figs. 9a, c). The Cor and NSE for the CMFD daily
average temperature were also similar to our estimated daily maximum and minimum temperatures
(Table 2). By contrast, the MAE and RMSE for the CMFD daily average temperature were 1.12 °C
and 1.64 °C, respectively, which were greater than for our estimated daily maximum and minimum
temperatures (Table 2). The MAEs of daily maximum and minimum temperature for our dataset
were 1.07 °C and 1.08 °C respectively; and the RMSEs of daily maximum and minimum
temperature for our dataset were 1.63 °C and 1.54 °C, respectively, between 1979 and 2018 (Table
2). The $R^2$, Cor, NSE, MAE, and RMSE for the CLDAS daily maximum temperatures were 0.91,
0.95, 0.90, 2.54 °C, and 3.63 °C, respectively. Accuracy clearly improved for our daily maximum
temperature, and the corresponding metrics were 0.98, 0.99, 0.98, 1.10 °C, and 1.73 °C (Figs. 10a,
b; Table 3). The MAE and RMSE for the CLDAS daily minimum temperature were clearly higher
than our estimates for daily minimum temperature, and the $R^2$, Cor, and NSE for daily minimum
temperature in our dataset were higher than those for the CLDAS daily minimum temperature (Figs.
10c, d; Table 3), thus indicating that the accuracy of our daily minimum temperature estimates was
superior to that of the CLDAS daily minimum temperature product. Compared with the ISIMIP3a,

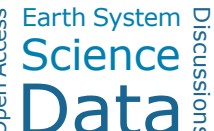

the $R^2$, Cor, and NSE of daily maximum and minimum temperature in our dataset are always higher
and the MAE and RMSE of these are always smaller (Figs. 11 a–d; Table 4).
The $R^2$ value for our estimated daily precipitation clearly improved compared to the other
three datasets, especially the ISIMIP3a and CLDAS dataset (Figs. 9d, e; Figs. 10e, f; Figs. 11e, f).
The Cor and NSE for the CMFD daily precipitation were obviously smaller than those for our
dataset, and the RMSE for CMFD daily precipitation were greater than those for our dataset (Table
2). During 2017–2019, the Cor, NSE, MAE, and RMSE for our estimated daily precipitation were
0.84, 0.70, 1.42 mm, and 4.93 mm, respectively, and the corresponding values for the CLDAS daily
precipitation changed to 0.58, 0.28, 2.36 mm, and 7.67 mm, respectively (Table 3). During 1961–
2016, the Cor, NSE, MAE, and RMSE for our estimated daily precipitation were 0.84, 0.70, 1.30
mm, and 4.78 mm, respectively, and the corresponding values for the ISIMIP3a daily precipitation
changed to 0.48, 0.14, 2.75 mm, and 8.10 mm, respectively (Table 4). Thus, the daily precipitation
accuracy of our dataset was generally higher than that of CMFD, CLDAS and ISIMIP3a.

## 349  5 Data availability

The HRLT dataset includes daily maximum temperature, minimum temperature, and
precipitation at a 1 km spatial resolution across China from January 1961 to December 2019. The
datasets are publicly available in NetCDF format at https://doi.org/10.1594/PANGAEA.941329
(Qin and Zhang, 2022).

## 354  6 Conclusions

The result of this study is a high-resolution (1 km) daily gridded maximum temperature,
minimum temperature and precipitation dataset across China for the long-term (1961–2019)



(HRLT). The HRLT dataset shows an overall high correlation with the observations from
meteorological stations for daily maximum and minimum temperatures ($R^2$ was 0.98 and 0.99,
respectively; Cor were both 0.99; NSE was 0.98 and 0.99, respectively) and the errors were smaller
(MAE was 1.07 ℃ and 1.08 ℃, respectively; RMSE was 1.62 ℃ and 1.53 ℃, respectively).
Although the HRLT dataset showed that the daily precipitation accuracy was lower than the daily
temperature accuracy ($R^2$, Cor, NSE, MAE, and RMSE were 0.71, 0.84, 0.70, 1.30 mm, and 4.78
mm, respectively), the daily precipitation data in the HRLT dataset were more accurate and had a
finer spatial resolution compared to the other three existing datasets (CMFD, CLDAS and
ISIMIP3a). Furthermore, the accuracies for daily maximum and minimum temperatures and
precipitation were lower in the southwestern part of China, probably because of the complex
topography in that area compared to other areas. Calculation and interpolation by subregions may
solve this problem in future studies. The use of satellite data as an input covariate in future studies
will further improve the accuracy of the HRLT dataset, especially for precipitation. The HRLT
dataset will help identify future extreme climatic events and can be also used to improve process-
based models for prediction, adaptation, and mitigation strategies.

## Author contributions

Rongzhu Qin and Feng Zhang calculated the dataset, analyzed the results, and wrote the
manuscript; all other authors reviewed and revised the manuscript.

## Competing interests

The authors declare that they have no conflict of interest.



# Acknowledgements


This study was supported by the Second Tibetan Plateau Scientific Expedition and Research
(Grant No. 2019QZKK0305), National Natural Science Foundation of China (Grant No. 32071550),
and the "111" Programme (BP0719040). We thank the Supercomputing Center of Lanzhou
University for providing the computing requirements for this study. We are also grateful to the China
Meteorological Data Service Centre, National Tibetan Plateau Third Pole Environment Data Center,
Potsdam Institute for Climate Impact Research and the International Institute for Applied Systems
Analysis for contributing datasets to the study. We are also thanking Dr. Marianne Rehage from
PANGAEA for processing and publishing the dataset.

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



**Table 1** Summary of the accuracies for the HRLT datasets using data from the meteorological stations

| Variable | MAE | RMSE | Cor | NSE | N | Period |
|---|---|---|---|---|---|---|
| Maximum temperature (℃) | 1.07 | 1.62 | 0.99 | 0.98 | 14731830 | 1961–2019 |
| Minimum temperature (℃) | 1.08 | 1.53 | 0.99 | 0.99 | 14730410 | 1961–2019 |
| Precipitation (mm) | 1.30 | 4.78 | 0.84 | 0.70 | 14730380 | 1961–2019 |

MAE, RMSE, Cor, and NSE are the mean absolute error, root mean square error, Pearson's correlation coefficient, and Nash-Sutcliffe modeling efficiency, respectively. N is the number of observations and Period is the beginning to end years of the data.



**Table 2** Comparison of accuracies for the HRLT and CMFD datasets using data from the meteorological stations

| Variable | Dataset | MAE | RMSE | Cor | NSE | N | Period |
|---|---|---|---|---|---|---|---|
| Maximum temperature (℃) | HRLT | 1.07 | 1.63 | 0.99 | 0.98 | 9969602 | 1979–2018 |
| Minimum temperature (℃) | HRLT | 1.08 | 1.54 | 0.99 | 0.99 | 9969602 | 1979–2018 |
| Average temperature (℃) | CMFD | 1.12 | 1.64 | 0.99 | 0.98 | 9969602 | 1979–2018 |
| Precipitation (mm) | HRLT | 1.30 | 4.73 | 0.84 | 0.71 | 9968784 | 1979–2018 |
| | CMFD | 1.30 | 5.85 | 0.75 | 0.55 | 9968784 | 1979–2018 |

MAE, RMSE, Cor, and NSE are the mean absolute error, root mean square error, Pearson's correlation coefficient, and Nash-Sutcliffe modeling efficiency, respectively. N is the number of observations and Period is the beginning to end years of the data.





**Table 3** Comparison of accuracies for the HRLT and the CLDAS datasets using data from the meteorological stations

| Variable | Dataset | MAE | RMSE | Cor | NSE | N | Period |
|---|---|---|---|---|---|---|---|
| Maximum | HRLT | 1.10 | 1.73 | 0.99 | 0.98 | 686653 | 2017–2019 |
| temperature (℃) | CLDAS | 2.54 | 3.63 | 0.95 | 0.90 | 686653 | 2017–2019 |
| | | | | | | | |
| Minimum | HRLT | 1.14 | 1.65 | 0.99 | 0.98 | 686653 | 2017–2019 |
| temperature (℃) | CLDAS | 1.58 | 2.63 | 0.98 | 0.95 | 686653 | 2017–2019 |
| | | | | | | | |
| Precipitation | HRLT | 1.42 | 4.93 | 0.84 | 0.70 | 685936 | 2017–2019 |
| (mm) | CLDAS | 2.36 | 7.67 | 0.58 | 0.28 | 685936 | 2017–2019 |

MAE, RMSE, Cor, and NSE are the mean absolute error, root mean square error, Pearson's correlation coefficient, and Nash-Sutcliffe modeling efficiency, respectively. N is the number of observations and Period is the beginning to end years of the data.



**Table 4** Comparison of accuracies for the HRLT and the ISIMP3a datasets using data from the meteorological stations

| Variable | Dataset | MAE | RMSE | Cor | NSE | N | Period |
|---|---|---|---|---|---|---|---|
| Maximum | HRLT | 1.06 | 1.61 | 0.99 | 0.98 | 13973110 | 1961–2016 |
| temperature (℃) | ISIMP3a | 2.47 | 3.47 | 0.96 | 0.91 | 13973110 | 1961–2016 |
| Minimum | HRLT | 1.07 | 1.52 | 0.99 | 0.99 | 13971690 | 1961–2016 |
| temperature (℃) | ISIMP3a | 2.63 | 3.60 | 0.96 | 0.92 | 13971690 | 1961–2016 |
| Precipitation | HRLT | 1.30 | 4.78 | 0.84 | 0.70 | 13971680 | 1961–2016 |
| (mm) | ISIMP3a | 2.75 | 8.10 | 0.48 | 0.14 | 13971680 | 1961–2016 |

MAE, RMSE, Cor, and NSE are the mean absolute error, root mean square error, Pearson's correlation coefficient, and Nash-Sutcliffe modeling efficiency, respectively. N is the number of observations and Period is the beginning to end years of the data.



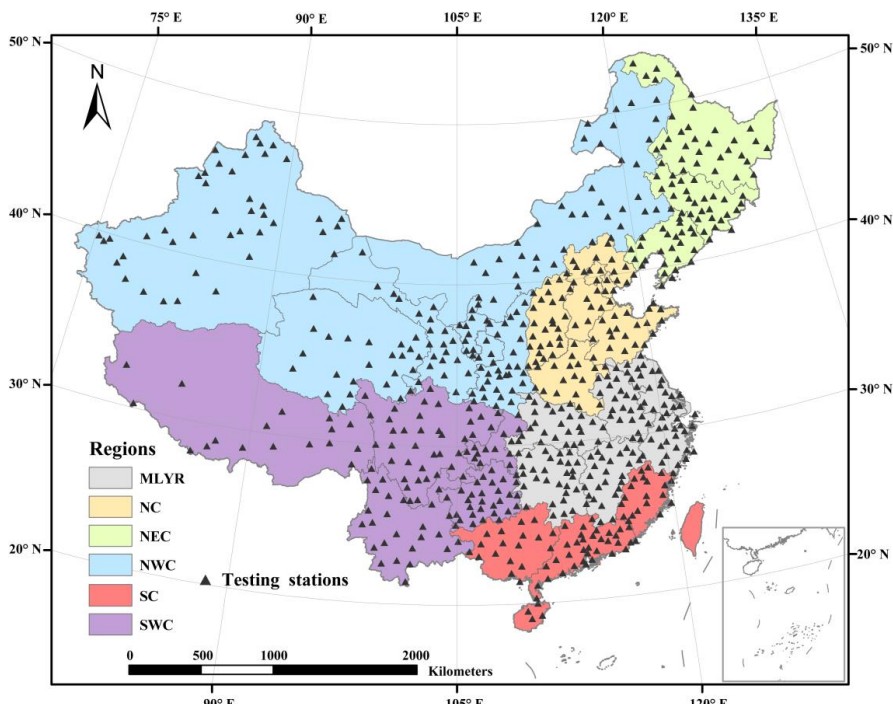

**Figure 1.** Regions and spatial distribution of the meteorological stations in China. MLYR, NC, NEC, NWC, SC, and SWC are the Middle and Lower reaches of the Yangtze River, North China, Northeast China, Northwest China, South China, and Southwest China, respectively. Note: meteorological stations data were missing for Taiwan Province.

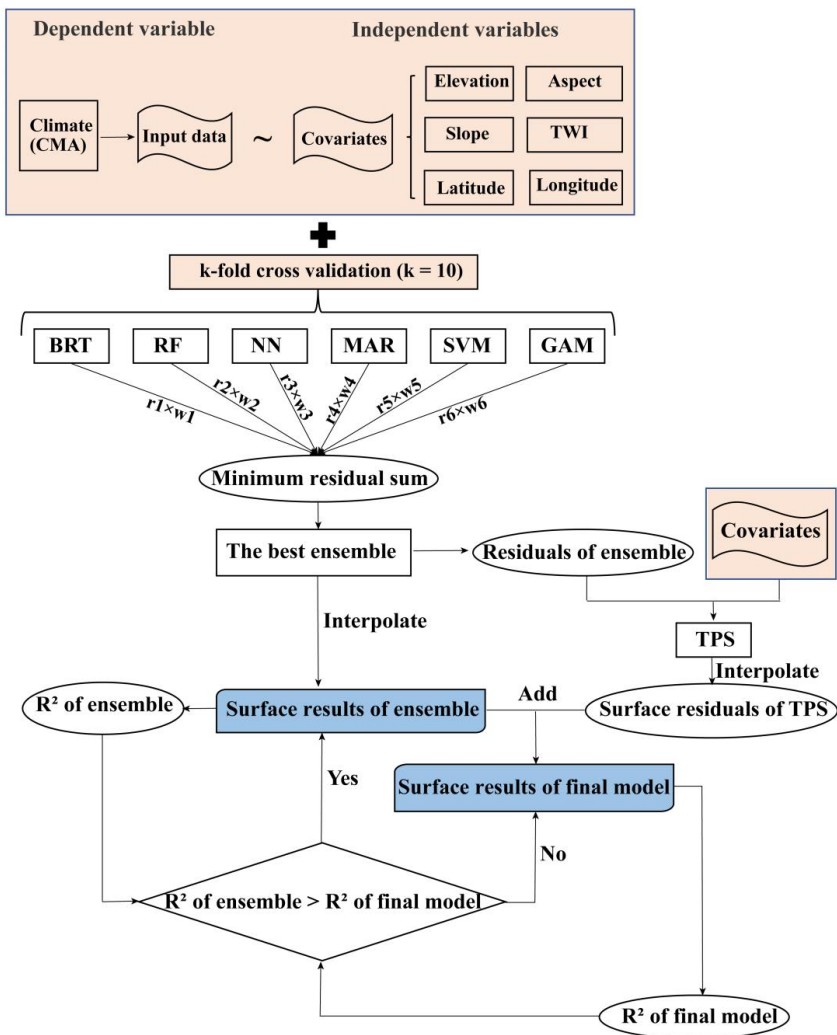

**Figure 2.** The process of spatial interpolation. The r1 to r6 are the residual error from each algorithm, respectively. The w1 to w6 are the weights of each algorithm, respectively. BRT, RF, NN, MAR, SVR, GAM and TPS are the boosted regression trees, random forests, neural networks, multivariate adaptive regression splines, support vector machines, the generalized additive model and thin-plate-smoothing splines, respectively. $R^2$ is the coefficient of determination between the estimated and observed values.



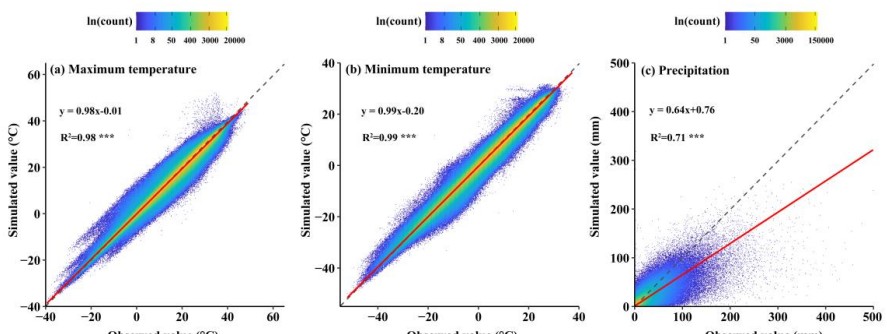

**Figure 3.** Scatter density plots of daily maximum and minimum temperatures and precipitation between estimated and observed values at meteorological stations were used to test the HRLT dataset. Dashed line is a line with slope 1 and the red line is a fitting between estimated and observed values. $R^2$ is the coefficient of determination between the estimated and observed values. *** asterisks indicate that the significance of the regression equation between the estimated and observed values was $p < 0.001$.

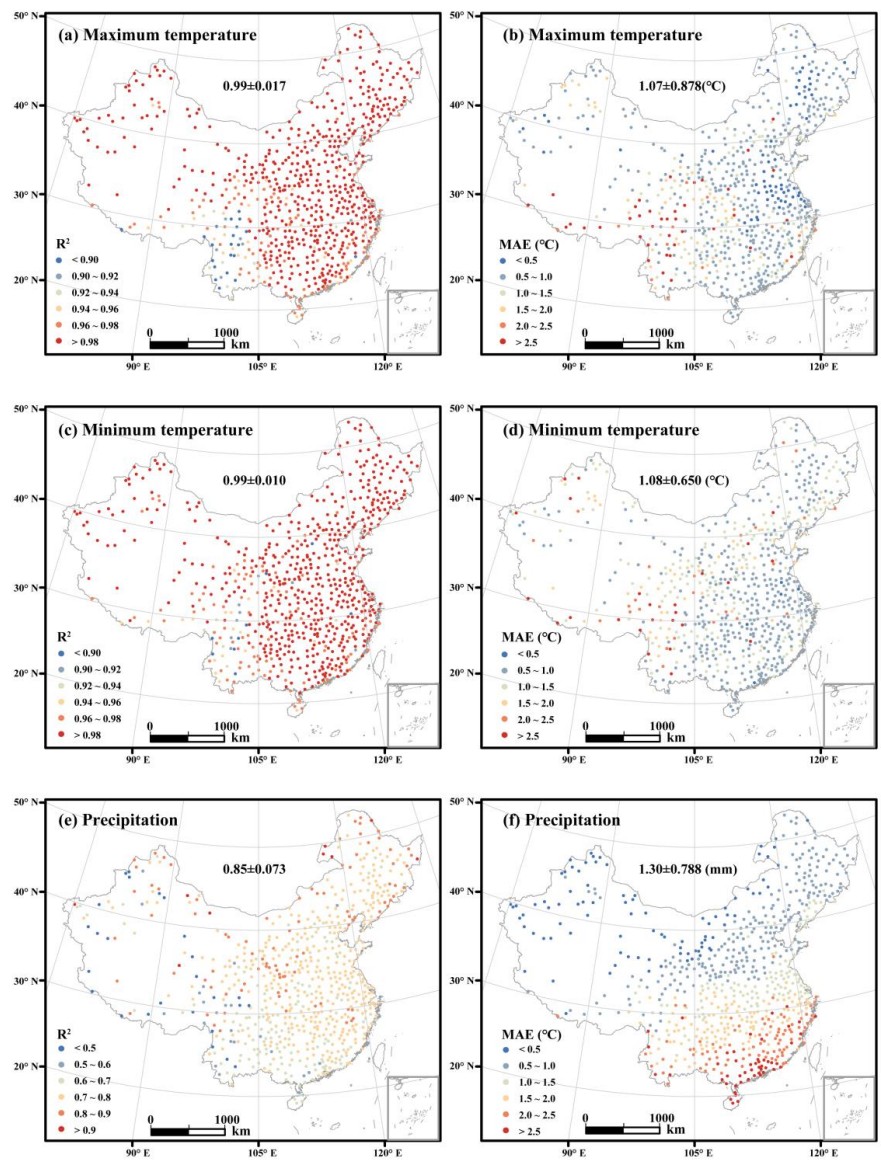

**Figure 4.** Spatial distribution of $R^2$ and MAE for daily maximum temperature, minimum temperature, and precipitation between 1961 and 2019. The value before the ± is the $R^2$ or MAE mean value and the value after the ± is the $R^2$ or MAE standard deviation for all meteorological stations.

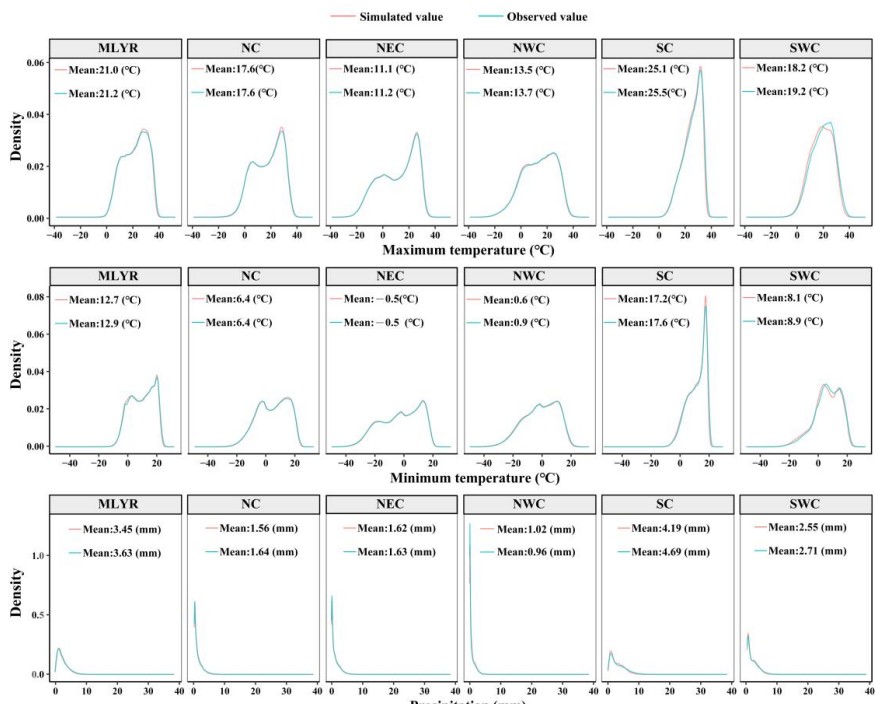

**Figure 5.** Comparisons of the density distribution between the estimated value in our dataset and the observed values from meteorological stations for daily maximum temperature, minimum temperature, and precipitation in the different regions from 1961 to 2019.

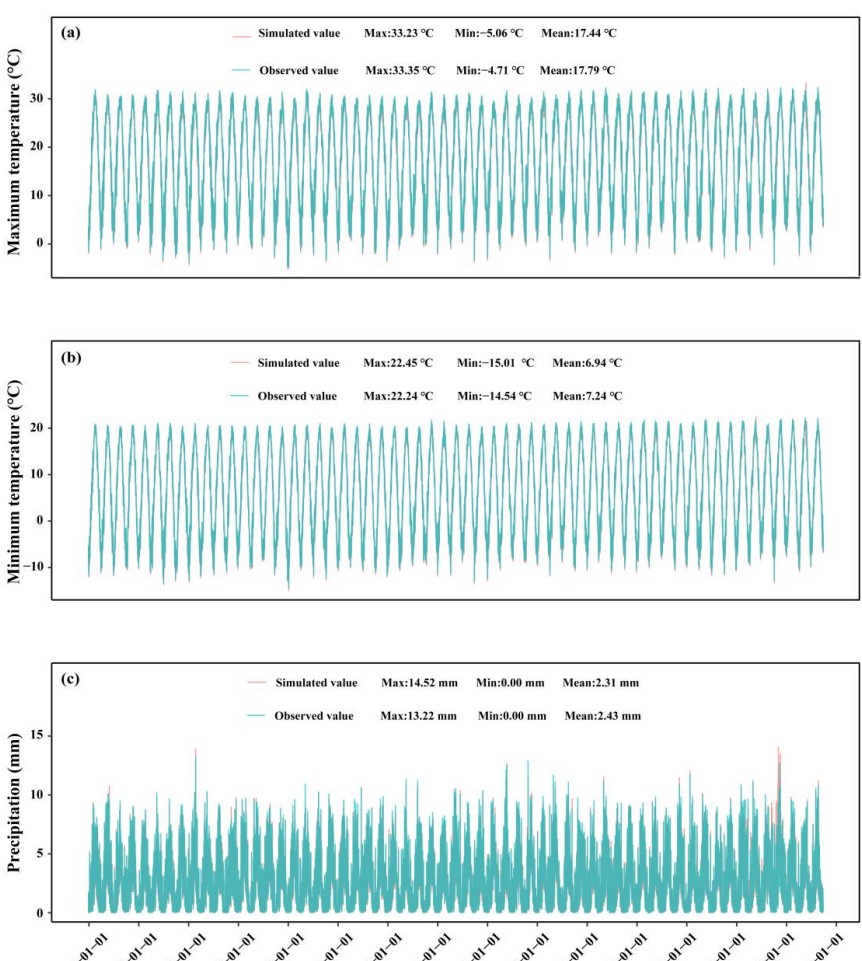

**Figure 6.** Comparisons of the daily changes between the estimated and observed values for daily maximum temperature, minimum temperature, and precipitation from January 1, 1961 to December 31, 2019 over all meteorological stations.

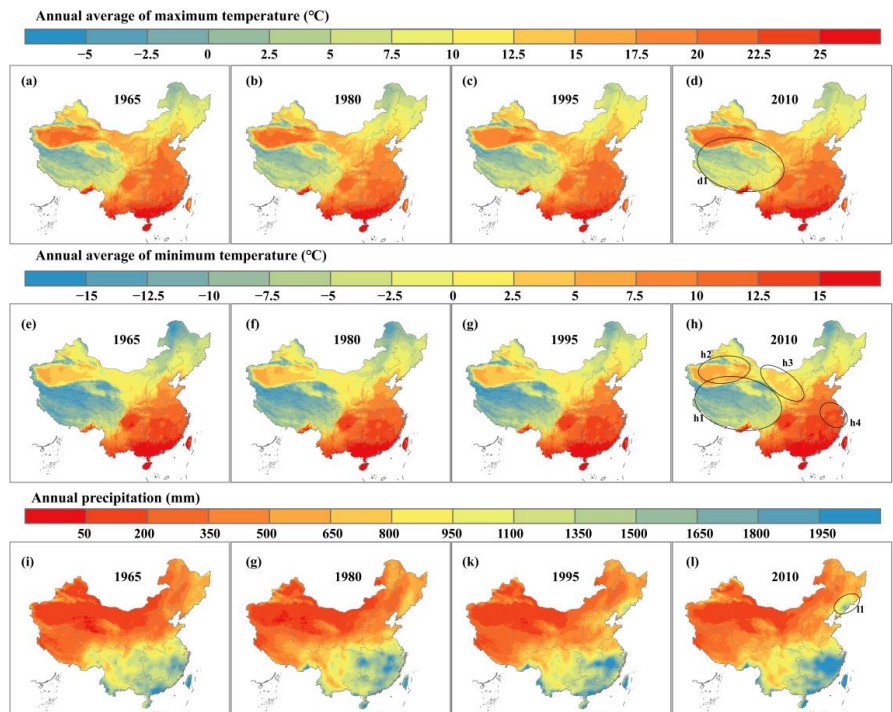

**Figure 7.** Spatial distributions of annual average values for daily maximum and minimum temperatures, and the spatial distribution of annual precipitation in 1965, 1980, 1990, and 2010. The ellipse regions are where the change is most visible.

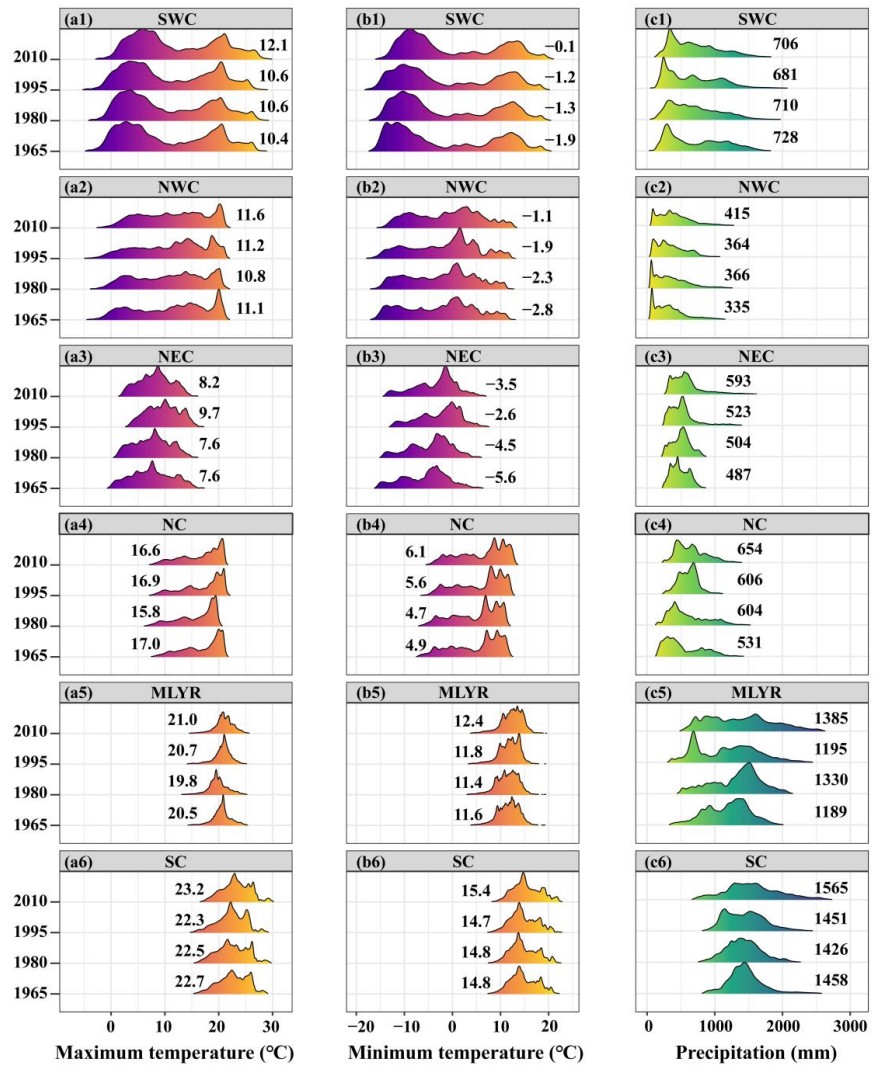

**Figure 8.** Density distributions of annual average values for daily maximum and minimum temperatures, and annual precipitation across the different regions in 1965, 1980, 1990, and 2010. The value in the illustrations is the mean value.

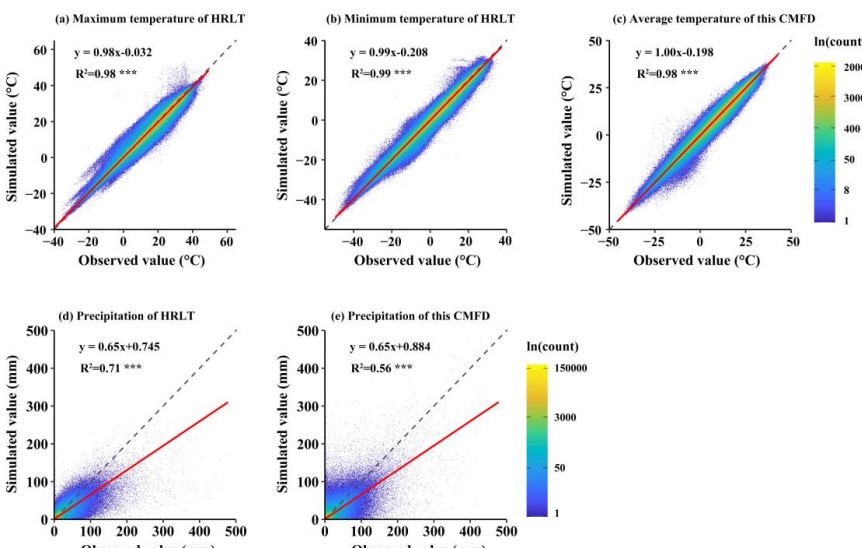

**Figure 9.** Scatter density plots of daily temperature and precipitation between the estimated and observed values at all meteorological stations (both training sets and testing sets) for the HRLT dataset and the CMFD dataset between 1979 and 2018. The dashed line is a line with slope 1 and the red line is a fitting between the estimated and observed values. $R^2$ is the coefficient of determination between the estimated and observed values. *** asterisks indicate that the significance of the regression equation between the estimated and observed values was $p < 0.001$.

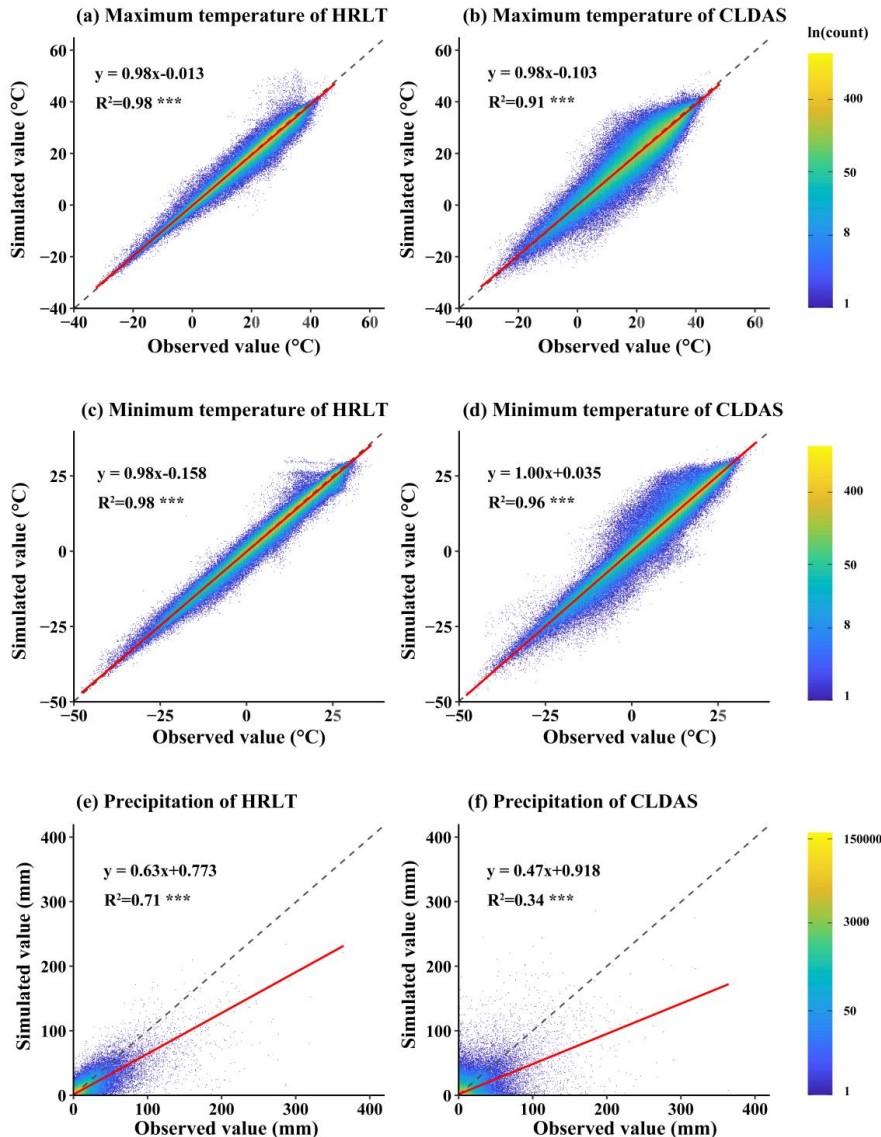

**Figure 10.** Scatter density plots of daily temperature and precipitation between the estimated and observed values from all meteorological stations (both training sets and testing sets) for our HRLT dataset and the CLDAS dataset between 2017 and 2019. Dashed line is a line with slope 1 and the red line is the fitting between the estimated and observed values. $R^2$ is the coefficient of determination between the estimated and observed values. *** asterisks indicate that the significance of the regression equation between the estimated and observed values was $p < 0.001$.

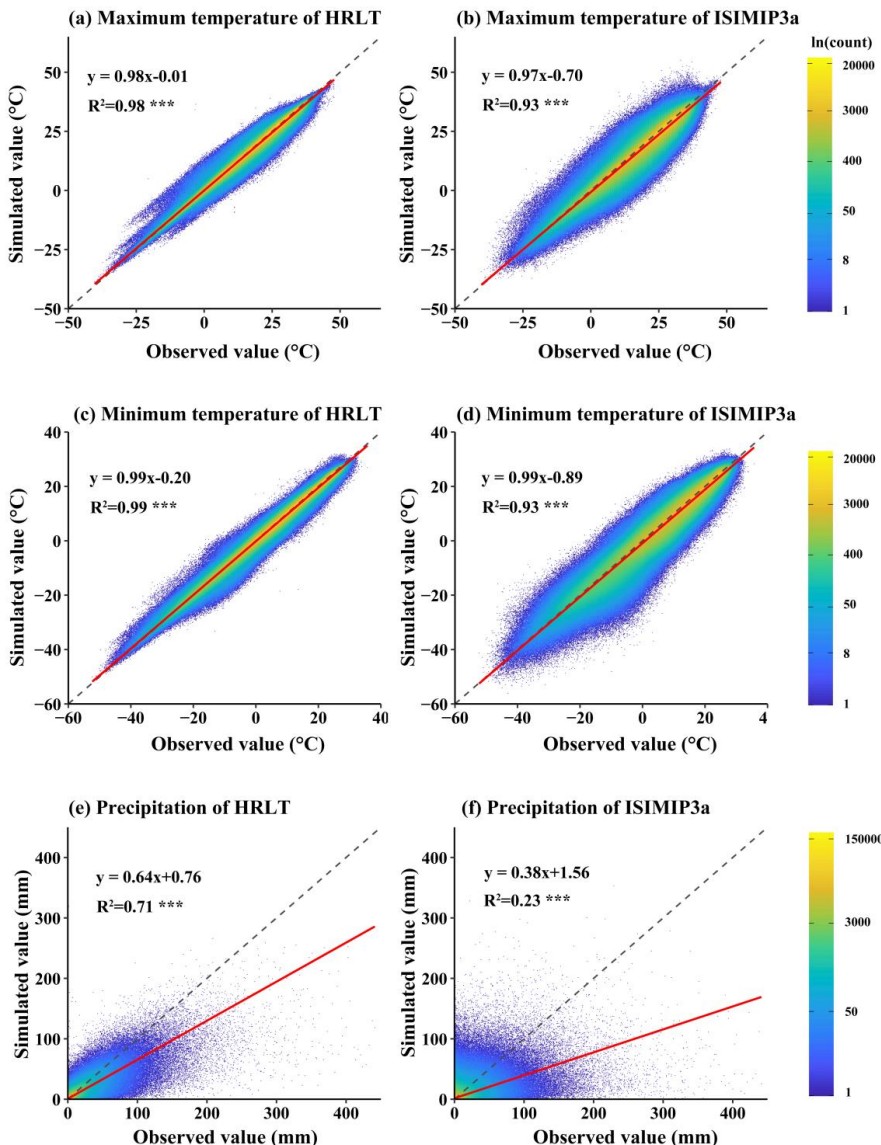

**Figure 11.** Scatter density plots of daily temperature and precipitation between the estimated and observed values from all meteorological stations (both training sets and testing sets) for our HRLT dataset and the ISIMIP3a dataset between 1961 and 2016. Dashed line is a line with slope 1 and the red line is the fitting between the estimated and observed values. $R^2$ is the coefficient of determination between the estimated and observed values. *** asterisks indicate that the significance of the regression equation between the estimated and observed values was $p < 0.001$.