# Peer review of "HRLT: A high-resolution (1 day, 1 km) and long-term"

_Earth System Science Data, 2022_

## Referee Comment (RC1)

**Review of the manuscript titled "HRLT: A high-resolution (1 day, 1 km) and long-term (1961–2019) gridded dataset for temperature and precipitation across China".**

**General comments:**

This study produced a new daily maximum temperature, minimum temperature, and precipitation dataset spanning 1961-2019 in spatial resolution of 1km in China by various machine learning and traditional methods. The observation data from meteorological stations were used to evaluate this dataset and the other three existing datasets, the result showed this dataset with high accuracy. This study involved huge computation with interpolation, and the method, which will be helpful for similar studies. Overall, the manuscript was well written and easy to follow. The method is very reliable, the dataset is solid and it will have valuable contributions in the fields of ecology, remote sensing, hydrology, and meteorological science. This manuscript and dataset have the potential to be a highly cited work in the future. In my opinion, the MS can be accepted for publication after a medium revision.

**Specific comments:**

1. For all reference links in the manuscript, the last access date should be added and the font color should be uniform (blue). For the simplicity and clarity of the manuscript, it is recommended to delete the reference links of temperature, precipitation, and meteorological stations in subsection 2.1 (Lines: 88-98), and uniformly use the link: (https://data.cma.cn/, last access: date).

2. There are many grammar problems (such as Line 31: Replace the "is" with "was"; Line 62: Replace the "grid" with "gridded"). Please modify these problems in the manuscript accordingly.

3. Line 30: More than one machine learning method is used in this manuscript, so it is recommended to change "machine learning" to "machine learning methods". This also works for line 80.

4. Figure 1: The legend representation as "Meteorological stations" is more appropriate than "Testing stations".

5. Line 106: The equation should be re-expressed on a new line, such as the equation line 238.

6. Lines 108-110: The expression is not clear, please re-write the sentence. Moreover, you should elaborate on which observed meteorological station data you used.

7. Line 134, please remove repeated "was"

8. Line 142: use the abbreviation TPS for thin-plate-smoothing splines

9. Line 144: Re-write the subheading ("The methods" to "The interpolation methods")

10. Line 147-148: add implemented after used to.

11. Line 153: Remote sensing uses low case R

12. Line 160: Remove repeated "The".

13. Line 324: Remove "for"

14. Figure 2: Please add the full spelling of TWI in the figure comments.

15. Figure 3&5: The values (R2/Mean) appear to be incomplete. please regenerate these figures.

16. Lines 265-268: please try to add the reference.

17. Figure 5&6: Please add the explanation of Min, Max, and Mean in the figure comments.

18. Figure 8b5: There seems to be an extra vertical white line, please delete it.

19. Lines 419-420: Reference missing page number, please add it.

20. Lines 476-480: The year of reference is incorrect. Please modify it

---

## Author Response (AR1)

Ref. No.: ESSD-2022-79

Title: "HRLT: A high-resolution (1 day, 1 km) and long-term (1961–2019) gridded dataset for temperature and precipitation across China"

Dear reviewer#1,

We are truly grateful for the constructive comments and thoughtful suggestions you provided. Based on these comments and suggestions, we have made careful modifications to the original manuscript. We hope the revised manuscript will meet journal's standard. Below you will find our point-by-point responses to the reviewer' comments/ questions in BLUE. Please let us know if you have any questions.

Yours sincerely,

Feng Zhang

**General comments:** This study produced a new daily maximum temperature, minimum temperature, and precipitation dataset spanning 1961-2019 in spatial resolution of 1km in China by various machine learning and traditional methods. The observation data from meteorological stations were used to evaluate this dataset and the other three existing datasets, the result showed this dataset with high accuracy. This study involved huge computation with interpolation, and the method, which will be helpful for similar studies. Overall, the manuscript was well written and easy to follow. The method is very reliable, the dataset is solid and it will have valuable contributions in the fields of ecology, remote sensing, hydrology, and meteorological science. This manuscript and dataset have the potential to be a highly cited work in the future. In my opinion, the MS can be accepted for publication after a medium revision.

**Specific comments:**

1. For all reference links in the manuscript, the last access date should be added and the font color should be uniformed (blue). For the simplicity and clarity of the manuscript, it is recommended to delete the reference links of temperature, precipitation, and meteorological stations in subsection 2.1 (Lines: 88-98), and uniformly use the link: (https://data.cma.cn/, last access: date).

   Agreed, the reference links have been appended and revised.

2. There are many grammar problems (such as Line 31: Replace the "is" with "was"; Line 62: Replace the "grid" with "gridded"). Please modify these problems in the manuscript

accordingly.

Agreed, we have fixed the grammar problems.

3. Line 30: More than one machine learning method is used in this manuscript, so it is recommended to change "machine learning" to "machine learning methods". This also works on line 80.

Agreed, we have replaced the "machine learning" with "machine learning methods".

4. Figure 1: The legend representation as "Meteorological stations" is more appropriate than "Testing stations".

Agreed, we have replaced the "Testing stations" with "Meteorological stations":

[Figure]

5. Line 106: The equation should be re-expressed on a new line, such as the equation line 238.

The TWI is formulated as follow:

$$\text{TWI} = ln(\frac{\text{SCA}}{\text{tan(Slope)}}) \qquad (1)$$

where TWI and SCA is topographic wetness index and specific catchment area, respectively.

6. Lines 108-110: The expression is not clear, please re-write the sentence. Moreover, you should elaborate on which observed meteorological station data you used.

We used observed data from meteorological stations (Fig. 1) to evaluate our dataset and the existing three daily datasets, then the accuracy of the existing three daily datasets was compared to that of our dataset, respectively.

7. Line 134, please remove repeated "was"

Agreed.

8. Line 142: use the abbreviation TPS for thin-plate-smoothing splines

Agreed.

9. Line 144: Re-write the subheading ("The methods" to "The interpolation methods")

We have changed the "4.3 The methods" to "4.3 The interpolation methods"

10. Line 147-148: add implemented after used to.

Agreed.

11. Line 153: Remote sensing uses low case R

Agreed.

12. Line 160: Remove repeated "The"

Agreed.

13. Line 324: Remove "for"

Agreed.

14. Figure 2: Please add the full spelling of TWI in the figure comments.

The TWI is topographic wetness index and it has been added in figure 2 comments

15. Figure 3&5: The values ($R^2$/Mean) appear to be incomplete. please regenerate these figures.

We have regenerated these figures:

[Figure]

[Figure]

16. Lines 265-268: please try to add reference.

The reference (Qin, et al., 2022) of different regions has been supplemented:

Qin, R., F. Zhang, C. Yu, Q. Zhang, J. Qi, and F.-m. Li: Contributions made by rain-fed potato with mulching to food security in China, European Journal of Agronomy, 133, 126435, https://doi.org/10.1016/j.eja.2021.126435, 2022.

17. Figure 5&6: Please add the explain of Min, Max and Mean in the figure comments.

Agreed, they have been added in figure 5&6 comments.

18. Figure 8b5: There seems to be an extra vertical white line, please delete it.

This may be due to discontinuous data distribution.

19. Lines 419-420: Reference missing page number, please add.

Friedman, J. H. and Roosen, C. B.: An introduction to multivariate adaptive regression splines, 3,192-217, https://doi.org/10.1177/096228029500400303, 1995.

20. Lines 476-480: The year of reference is incorrect. Please modify it

Merino, A., Guerrero-Higueras, A. M., López, L., Gascón, E., Sánchez, J. L., Lorente, J. M., Marcos, J. L., Matía, P., Ortiz de Galisteo, J. P., Nafría, D., Fernández-González, S.,

Weigand, R., Hermida, L., and García-Ortega, E.: Development of tools for evaluating rainfall estimation models in real- time using the Integrated Meteorological Observation Network in Castilla y León (Spain), 2014.

Ref. No.: ESSD-2022-79

Title:"HRLT: A high-resolution (1 day, 1 km) and long-term (1961–2019) gridded dataset for temperature and precipitation across China"

Dear reviewer#2,

We are truly grateful for the constructive comments and thoughtful suggestions you provided. Based on these comments and suggestions, we have made careful modifications to the original manuscript. We hope the revised manuscript will meet journal's standard. Below you will find our point-by-point responses to the reviewer' comments/ questions in BLUE, and the changed text of the manuscript in RED. Please let us know if you have any questions.

Yours sincerely,

Feng Zhang

**General comments:** This paper introduces a new high resolution dataset of surface temperature and precipitation across China over a long term period. The data is obtained from original coarse (0.5°) resolution meteorological observations and downscaled to 1km using machine learning techniques. The algorithm employs a suite of techniques and the most performing is retained in the final estimate. A validation and intercomparison is conducted using station data and shows the improved score of the present dataset with respect to similar datasets already available. While the temperature downscaling is already very good in all the products, the improvements here are only marginal but more significant for precipitation. Yet even if improved the precipitation remains less well downscaled than temperature in a significant manner. A trend analysis is offered as an illustration of the interest of the dataset.

Overall the paper is clearly written and provide a very complete perspective on the dataset. The algorithm and the input data are well documented so are the intercomparison products. As such the paper is a good realization of a "data paper" and is very well suited for ESSD. I nevertheless have some remarks below that should be adressed prior to publication.

**Specific comments:**

(1) Figures 5 and 6 are unreadable. Please make these two figures more clear. For instance I suggest to plot the difference between the two variables in the hope it will show more their small departure that the currently useless figures.

Agreed, we showed the cumulative distribution functions of difference between the estimated

and observed values for three variables and represent the original figures 5 and 6 as the new Figure 6.

[Figure]

**Figure 6.** Cumulative distribution functions (CDF) of difference between the estimated and observed values for three variables in all meteorological stations from 1961 to 2020. μ is the mean and σ is the standard deviation. MLYR, NC, NEC, NWC, SC, and SWC are the Middle and Lower reaches of the Yangtze River, North China, Northeast China, Northwest China, South China, and Southwest China, respectively.

The cumulative distribution functions curve trend of difference between the estimated and observed values was always similar for daily maximum temperature, minimum temperature, and precipitation in the six regions, as well as in whole China. The daily maximum and minimum temperatures were all underestimated in the MLYR, NEC, NWC, SC, and SWC (Fig. 5a). The daily minimum temperatures were all underestimated in the MLYR, NC, NWC, SC, and SWC (Fig. 5b). For both daily maximum and minimum temperatures, the lowest average difference between the simulated and observed values occurred in NC and NEC, while the greatest difference occurred in SWC (Figs. 5a,b). Except in the NWC region, the average difference between simulated and observed values for daily precipitation was less than 0 mm in the other regions (Fig. 5c). The largest difference between simulated and observed averages for daily precipitation occurred in the SC region, with a value of 0.49 mm (Fig. 5c). Across whole China, the average difference between simulated and observed values for daily maximum temperature, minimum temperature, and precipitation was 0.36 ℃, 0.30 ℃ and 0.12 mm, respectively.

(2) Also I think there is a geographically distributed bias in the performance of the new products that is barely mentioned and not enough discussed. In particular for precipitation where the correlation map (Figure 4 e) shows a west-east gradient in the scores that is different

from the north-south gradient in the MAE map (figure 4f). This should be discussed in more depths and possible, if not definitive, explanations for such a pattern to be proposed.

Thank you very much for the suggestion. In sections 4.1, we have mentioned the geographically distributed bias in the performance of this dataset . For precipitation, the pattern has been discussed.

[Figure]

**Figure 5.** The relationship between latitude and MAE of daily precipitation. Illustration indicates the relationship between rainfall frequency above light rainfall and MAE of daily precipitation. MAE is the mean absolute error, Cor is Pearson's correlation coefficient, Rain frequency is rainfall frequency above light rainfall, which is defined as a daily rainfall from 0 to 4 mm (Alpert et al., 2002)

For precipitation where the $R^2$ map (Fig. 4e) shows a west-east gradient in the scores that is different from the north-south gradient in the MAE map (Fig. 4f). There are fewer meteorological observation stations in the western region than in the eastern region, which may lead to the subtle east-west gradient of the $R^2$ value for daily precipitation. The obvious north-south gradient for MAE of daily precipitation could be caused by the rainfall frequency (Fig. 4f, Fig. 5), the MAE of monthly precipitation in China from other study showed a similar

pattern (Peng et al., 2019). Rainfall frequency above light rainfall, which is defined as a daily rainfall from 0 to 4 mm (Alpert et al., 2002), is strongly correlated with the MAE of daily precipitation (illustration in Fig. 5), so that the MAE of daily precipitation in the southern region with higher rainfall frequency is larger than that in the northern region with lower rainfall frequency.

Alpert, P., T. Ben-Gai, A. Baharad, Y. Benjamini, D. Yekutieli, M. Colacino, L. Diodato et al. The paradoxical increase of Mediterranean extreme daily rainfall in spite of decrease in total values, Geophysical research letters 29(11):31-1, https://doi.org/10.1029/2001GL013554, 2002.

Peng, S., Y. Ding, W. Liu, and Z. Li. 1 km monthly temperature and precipitation dataset for China from 1901 to 2017. Earth Syst. Sci. Data 11:1931-1946, https://doi.org/10.5194/essd-11-1931-2019, 2019.

(3) In the final sentence of the abstract (line 45), the authors state that such a data is fit for various studies especially for extreme weather related studies. This last statement is not supported by the paper and should be removed. In particular in light of the still weak, even if improved, performances for precipitation for which the more intense rain rates are not well reproduced by the gridded dataset.

Agreed, the final sentence of the abstract have been removed.

On a smaller note I would like to ask for more details on the rain-gauges dataset, in particular about the under catch corrections (if any) that is known to influence strongly the rain gauges estimates and likely the interpolation procedure.

The observed rain-gauges dataset was obtained from the China Meteorological Data Service Center. The rain gauges are sampled at a frequency of once per minute, with rainfall units in millimeters rounded to one decimal point. When the rainfall is less than 0.1mm/min, it will be regarded as no rainfall, which is 0 mm. More detailed information was described in the China's National standard "Specifications for surface meteorological observation-Precipitation (GB/T 3528-2017)".

Overall I support publication of the paper once the major items above are adressed.

miscellaneous: I suggest to add "surface" to the title

"gridded dataset for temperature and precipitation across China"-> "gridded dataset for surface temperature and precipitation across China".

Agreed, we have added "surface" to the title.

HRLT: A high-resolution (1 day, 1 km) and long-term (1961–2019) gridded dataset for surface temperature and precipitation across China

Dear community:

Thank you very much for your helpful comments to improve this manuscript.

**Comments:** This manuscript produced a downscaled temperature and rainfall dataset by comprehensive methods. As the author points out, such high-resolution dataset is very valuable for research in related fields, especially revealing the effects of extreme temperature and precipitation events on plant growth, etc.

However, I am more concerned about how to interpolate data in mountainous regions (such as the Qinghai-Tibet Plateau), as we all know that the topography in these areas is complex, and the interpolation accuracy in mountainous regions may be relatively low compared to other regions. Did the authors perform different methods in these regions to ensure high accuracy? I'm interested in whether this dataset has improved interpolation accuracy in mountainous regions compared to other existing datasets.

In addition, I found that some figures in the manuscript are of poor resolution and need further improvement. ion, the MS can be accepted for publication after a medium revision.

**Response:** In fact, we did not perform partition processing during the interpolation process and the results show that the accuracies were lower in the Southweste region (where there are many mountain distribution areas). Calculation and interpolation by subregions may solve this problem in future studies. Analysis of the dataset in the Southwest region shows that the accuracy in this region is indeed improved compared to other existing datasets, especially rainfall ( Figs. CC1-3). However, the improvement in southweste region is almost consistent with that in whole China. All figures in this MS was improved.

[Figure]

**Figure CC1.** Scatter density plots of daily temperature and precipitation between the estimated and observed values at meteorological stations in Southweste region for the HRLT dataset and the CMFD dataset between 1979 and 2018. The dashed line is a line with slope 1 and the red line is a fitting between the estimated and observed values. $R^2$ is the coefficient of determination between the estimated and observed values. MAE, RMSE, Cor, and NSE are the mean absolute error, root mean square error, Pearson's correlation coefficient, and Nash-Sutcliffe modeling efficiency, respectively.

[Figure]

**Figure CC2.** Scatter density plots of daily temperature and precipitation between the estimated and observed values at meteorological stations in Southweste region for the HRLT dataset and the CLDAS dataset between 2017 and 2019. The dashed line is a line with slope 1 and the red line is a fitting between the estimated and observed values. $R^2$ is the coefficient of determination between the estimated and observed values. MAE, RMSE, Cor, and NSE are the mean absolute error, root mean square error, Pearson's correlation coefficient, and Nash-Sutcliffe modeling efficiency, respectively.

[Figure]

**Figure CC3.** Scatter density plots of daily temperature and precipitation between the estimated and observed values at meteorological stations in Southweste region for the HRLT dataset and the ISIMISP3a dataset between 1961 and 2016. The dashed line is a line with slope 1 and the red line is a fitting between the estimated and observed values. $R^2$ is the coefficient of determination between the estimated and observed values. MAE, RMSE, Cor, and NSE are the mean absolute error, root mean square error, Pearson's correlation coefficient, and Nash-Sutcliffe modeling efficiency, respectively.